# Dipeptidyl Peptidase-4 Inhibitor Sitagliptin Phosphate Accelerates Cellular Cholesterol Efflux in THP-1 Cells

**DOI:** 10.3390/biom13020228

**Published:** 2023-01-24

**Authors:** Tomohiro Komatsu, Satomi Abe, Shihoko Nakashima, Kei Sasaki, Yasuki Higaki, Keijiro Saku, Shin-ichiro Miura, Yoshinari Uehara

**Affiliations:** 1Research Institute for Physical Activity, Fukuoka University, 8-19-1 Nanakuma, Johnan-ku, Fukuoka 814-0180, Japan; 2Center for Preventive, Anti-Aging and Regenerative Medicine, Fukuoka University Hospital, 7-45-1 Nanakuma, Johnan-ku, Fukuoka 814-0180, Japan; 3Faculty of Sports and Health Science, Fukuoka University, 8-19-1 Nanakuma, Johnan-ku, Fukuoka 814-0180, Japan; 4Department of Cardiology, Fukuoka University Hospital, 7-45-1 Nanakuma, Johnan-ku, Fukuoka 814-0180, Japan

**Keywords:** HDL, cholesterol efflux, DPP-4 inhibitor, GLP-1, ABCA1, apoA1

## Abstract

Cholesterol efflux is a major atheroprotective function of high-density lipoproteins (HDLs) which removes cholesterol from the foam cells of lipid-rich plaques in Type 2 diabetes. The dipeptidyl peptidase-4 (DPP-4) inhibitor sitagliptin phosphate increases plasma glucagon-like peptide-1 (GLP-1) concentrations and is used to treat Type 2 diabetes. GLP-1 plays an important role in regulating insulin secretion and expression via the GLP-1 receptor (GLP-1R), which is expressed in pancreatic islets as well as freshly isolated human monocytes and THP-1 cells. Here, we identified a direct role of GLP-1 and DPP-4 inhibition in HDL function. Cholesterol efflux was measured in cultivated phorbol 12-myristate 13-acetate-treated THP-1 cells radiolabeled with ^3^H-cholesterol and stimulated with liver X receptor/retinoid X receptor agonists. Contrary to vildagliptin, sitagliptin phosphate together with GLP-1 significantly (*p* < 0.01) elevated apolipoprotein (apo)A1-mediated cholesterol efflux in a dose-dependent manner. The sitagliptin-induced increase in cholesterol efflux did not occur in the absence of GLP-1. In contrast, adenosine triphosphate-binding cassette transporter A1 (ABCA1) mRNA and protein expressions in the whole cell fraction were not changed by sitagliptin in the presence of GLP-1, although sitagliptin treatment significantly increased ABCA1 protein expression in the membrane fraction. Furthermore, the sitagliptin-induced, elevated efflux in the presence of GLP-1 was significantly decreased by a GLP-1R antagonist, an effect that was not observed with a protein kinase A inhibitor. To our knowledge, the present study reports for the first time that sitagliptin elevates cholesterol efflux in cultivated macrophages and may exert anti-atherosclerotic actions that are independent of improvements in glucose metabolism. Our results suggest that sitagliptin enhances HDL function by inducing a de novo HDL synthesis via cholesterol efflux.

## 1. Introduction

Reverse cholesterol transport (RCT) refers to the mechanism of cholesterol efflux, wherein cholesterol is extracted from atherosclerotic sites, transported to the liver, excreted into bile, and finally excreted via feces [1]. High-density lipoprotein (HDL) and apolipoprotein A1 (apoA1), a major component of HDL, play important roles in cholesterol efflux from macrophages, which is the first step in RCT: inhibiting atherosclerosis [2]. Adenosine triphosphate-binding cassette transporter A1 (ABCA1) membrane proteins transport free cholesterol and phospholipids from the membranous inner leaflet to the outer leaflet, and lipid-free or lipid-poor apoA1 subsequently takes up the transported cholesterol and phospholipids to form discoidal, nascent HDL [3]. In addition to having a role in removing cholesterol from foamy macrophages, ABCA1 is a key molecule in the formation of nascent HDL, an initial step in HDL metabolism. Previous studies found that the manipulated deletion of the ABCA1 gene exacerbates the development of atherosclerosis [4], while ABCA1 overexpression promotes RCT [5], suggesting that ABCA1 may play an essential role as an atherosclerosis suppressor in vivo. It is widely recognized that cyclic adenosine 3′,5′-monophosphate (cAMP) promotes apoA1-mediated cellular cholesterol efflux via activation of ABCA1 [6,7]. It has also been shown that the activation of protein kinase A (PKA) induces ABCA1 phosphorylation and promotes cholesterol efflux from fibroblasts [8].

In many populations, low blood levels of HDL cholesterol are a common component of metabolic syndrome, which is closely linked to being overweight or obese and having glucose intolerance or overt diabetes mellitus, hypertriglyceridemia, and hypertension, which individually contribute to the pathogenesis of atherosclerosis [9]. In addition, many individuals with low HDL cholesterol display elevated fasting-plasma-insulin levels, resistance to exogenous insulin in euglycemic clamp studies, and an increased risk of Type 2 diabetes [10,11,12]. 

Glucagon-like peptide-1 (GLP-1) is an incretin hormone produced as a post-translational product of proglucagon processing in intestinal cells. It plays an important physiological role in regulating insulin secretion and expression [13] and stimulates an insulin response via the GLP-1 receptor (GLP-1R) in pancreatic β-cells [14,15,16,17]. Previous studies showed that GLP-1R was expressed in a cell- and tissue-specific manner in human insulinomas [18], islets [19,20], gastric tumor cell lines [21], brains, and hearts [22]. Arakawa et al. demonstrated that GLP-1R was also expressed in murine macrophages, human monocytes, and pancreatic β-cells [23]. In addition, sitagliptin or vildagliptin treatment led to plaque regression in apolipoprotein (apo)E-deficient mice fed a high-fat diet [24,25,26] while increasing plaque stability and inhibiting monocyte migration [27]. These results suggest that GLP-1R may have direct anti-atherosclerotic effects independent of its ability to improve the glucose metabolism in monocytes/macrophages. It was recently reported that the clinical efficacy of GLP-1 agonists was associated with lower cardiovascular mortality [28]. The above clinical findings strongly imply that the GLP-1 pathway suppresses the development of atherosclerosis. The inhibition of dipeptidyl peptidase-4 (DPP-4) prolongs the action of incretin hormones, including GLP-1, by preventing their degradation [29], and it is also expected to be effective in reducing cardiovascular mortality.

Due to the fact that DPP-4 inhibitors showed anti-atherosclerotic effects in previous in vivo studies [24,25,27,30], the present study aims to clarify whether GLP-1 and an inhibition of DPP-4 increases ABCA1-mediated cholesterol efflux, a first step in the RCT process in monocytes/macrophages.

## 2. Materials and Methods

### 2.1. Materials

T0901317, 9-cis-retinoic acid, phorbol 12-myristate 13-acetate (PMA), GLP-1 (7–37), and exendin-(9–39) were purchased from Sigma (St Louis, MO, USA). Endotoxin, fatty-acid-free bovine serum albumin (BSA), and the PKA inhibitor 14–22 amide were purchased from Calbiochem (Merck KGaA, Darmstadt, Germany). Sitagliptin phosphate and vildagliptin were purchased from Toronto Research Chemicals Inc. (Toronto, ON, Canada). 

### 2.2. Cell Culture

THP-1 human monocytes (RIKEN, Tsukuba, Japan) were cultured in an RPMI-1640 medium containing 10% fetal bovine serum (Life Technologies Co, Carlsbad, CA, USA), 100 units/mL of penicillin G, and 100 μg/mL of streptomycin. THP-1 cells were treated with 10 μg/mL of PMA and 3.4% β-mercaptoethanol for 72 h prior to differentiation to macrophages [31]. After differentiation, the RPMI-1640 was used with with 4.5 g/L of high glucose for cholesterol efflux. For all experiments, cells were maintained in a serum-free medium containing 0.2% BSA supplemented with or without additives (5 μmol/L of T0901317 and 9-cis-retinoic acid or GLP-1). Sitagliptin phosphate, exendin-(9–39), and a PKA inhibitor were added 30 min prior to incubation with GLP-1. The concentrations of the DPP-4 inhibitor and GLP-1 in this study were selected in accordance with previous reports [24,30,32,33].

### 2.3. Cellular Cholesterol Efflux

THP-1 cells treated with PMA were radiolabeled with [1,2-^3^H] cholesterol (PerkinElmer, Waltham, MA, USA). The cellular cholesterol efflux was measured as described previously [31,34].

### 2.4. RNA Isolation and Quantitative, Real-Time Polymerase Chain Reaction (PCR)

Total RNA was extracted from cells in 9.6 cm^2^ dishes using the RiboPure kit (Life Technologies Co.) according to the manufacturer’s protocol. The RNA was reverse-transcribed, and the cDNA was amplified by PCR using the Transcriptor High Fidelity cDNA Synthesis kit (Roche Diagnosis GmbH, Mannheim, Germany). Gene-specific primers were used as follows: human ABCA1 5′–CCC TGT GGA ATG TAC CTA TGT G–3′ (forward), 5′–GAG GTG TCC CAA AGA TGC AA–3′ (reverse); human GAPDH 5′–CCC ATG TTC GTC ATG GGT GT–3′ (forward), and 5′–TGG TCA TGA GTC CTT CCA CGA TA–3′ (reverse) with annealing at 60 °C using 1.5 mmol/L of MgCl2. The gene expressions of human ABCA1 and GAPDH were quantified by real-time PCR using the Applied Biosystems GeneAmp 7500FAST sequence detection system (Applied Biosystems, Foster City, CA, USA). Amplification was detected using SYBR Green as a fluorogenic probe specific to double-stranded DNA, using a QuantiTect SYBR Green PCR kit (QIAGEN, Hilden, Germany). The threshold cycle, Ct, which was inversely correlated with the target mRNA levels, was measured as the cycle number at which the reported fluorescent emissions increased above a threshold level. Melting curves were recorded, and the size and specificity of the PCR products were confirmed using a 2% agarose gel (Sigma-Aldrich, St. Louis, MO, USA). Only reactions that produced a single band of the expected size were used for analysis. Data analysis was performed using the ΔΔCt method.

### 2.5. Western Blot Analysis

THP-1 cells from 55 cm^2^ dishes were washed with phosphate-buffered saline and collected in a 10 mmol/L HEPES (Sigma-Aldrich) with a complete protease inhibitor cocktail (Roche Molecular Biochemicals, Mannheim, Germany). Cell-membrane fractions were prepared in accordance with the methods described in previous reports for membrane ABCA1 detection [35,36]. Cell suspension was performed with three freeze–thaw cycles followed by ultrasonication and centrifugation at 900× *g* for 5 min, 4 ℃. The supernatant was then centrifuged at 20,000× *g* for 20 min, 4 ℃. After discarding the supernatant, the precipitated membrane was suspended in a RIPA lysis buffer (Sigma-Aldrich) supplemented with protease inhibitors. The extracted whole cells and membrane fractions were lysed in a buffer containing 50 mmol/L HEPES. The lysed cells were denatured at room temperature for 60 min in a buffer containing 100 mmol/L Tris (Sigma-Aldrich), 8% glycerol (Sigma-Aldrich), and 2% sodium dodecyl sulfate (SDS) (Sigma-Aldrich). An equal amount of protein was electrophoresed on 4–20% SDS-polyacrylamide gradient gels and then transferred onto a poly(vinylidene) fluoride (PVDF) microporous membrane (Millipore, Burlington, MA, USA) by electroblotting. After blocking with 5% skim milk, ABCA1 and β-actin were probed using mouse anti-human ABCA1 and rabbit β-actin antibodies (Abcam Inc., Cambridge, MA, USA), respectively. The immunoreaction was visualized after incubating the PVDF sheets with secondary horseradish-peroxidase-conjugated anti-mouse or anti-rabbit IgG antibodies (Bio Rad, Laboratories Inc., Hercules, CA, USA) using an ECL Western blotting detection reagent (GE healthcare, Amersham, Buckinghamshire, UK).

### 2.6. Statistical Analysis

Data were presented as the mean ± standard deviation (SD). The GraphPad Prism ver.8.43 software (GraphPad software LLC., San Diego, CA, USA) was used for all statistical analyses. Differences between groups were analyzed using the Kruskal–Wallis test with Dunn’s multiple comparison test analysis. A *p* value < 0.05 denoted a statistically significant difference.

## 3. Results

### 3.1. Sitagliptin Together with GLP-1 Significantly Increased Apoa1-Mediated Cholesterol Efflux

HDL removes excess cholesterol from lipid-rich macrophages and foam cells and plays an important role in anti-atherosclerosis. Similar to pancreatic β-cells [37], GLP-1R was abundantly expressed in freshly isolated human monocytes and THP-1 cells, which are human monocyte-like cells [23]. Therefore, we first examined the effects of GLP-1 on apoA1-mediated cholesterol efflux in THP-1 cells. Cellular cholesterol efflux was determined in THP-1 cells stimulated by a liver X receptor/retinoid X receptor (LXR/RXR) agonist to induce high levels of apoA1-mediated efflux. Cholesterol efflux in THP-1 cells was generally increased following apoA1 administration when compared to the control (BSA administration) (Figure 1A,B). Moreover, pretreatment with sitagliptin phosphate prior to incubation with GLP-1 significantly elevated apoA1-mediated cholesterol efflux in a dose-dependent manner (Figure 1A). However, the increased apoA1-mediated cholesterol efflux due to sitagliptin phosphate did not occur in the absence of GLP-1 (Figure 1B). Moreover, vildagliptin did not show the same effect for cholesterol efflux as sitagliptin (Figure 2).

### 3.2. ABCA1 Protein in Membrane Fraction, but Not Whole-Cell Fraction, Was Significantly Increased by Sitagliptin Phosphate

In contrast, the ABCA1 mRNA expression was not significantly increased following treatment with sitagliptin phosphate in the presence of GLP-1 (Figure 3A). Furthermore, sitagliptin phosphate did not affect the whole-cell expression of ABCA1 protein (Figure 3B). 

Interestingly, the ABCA1 protein expression was significantly elevated in the membrane fraction by sitagliptin phosphate in the presence of GLP-1 (Figure 4A), but not in its absence (Figure 4B). The elevated ABCA1 protein expression due to sitagliptin phosphate in the membrane fraction was slightly decreased following preincubation with the GLP-1R antagonist exendin-(9–39); however, this result was not statistically significant (Figure 4A). 

### 3.3. ApoA1-Mediated Cholesterol Efflux Was Mediated via GLP-1R Pathway

PKA was reported to accelerate the phosphorylation of the ABCA1 protein, promoting an apoA1-mediated cellular cholesterol efflux [8]. As is shown in Figure 5, a PKA inhibitor did not completely suppress the elevation of the apoA1-mediated cholesterol efflux by sitagliptin phosphate in the presence of GLP-1. However, a significant increase in apoA1-mediated cholesterol efflux due to sitagliptin phosphate was noted which, although not completely inhibited, was partially inhibited by preincubation with the GLP-1R antagonist exendin-(9–39), showing a similar tendency to that of the ABCA1 protein in the membrane fraction. 

## 4. Discussion

To our knowledge, the present study is the first to demonstrate that sitagliptin—but not vildagliptin—can elevate an apoA1-mediated cholesterol efflux in cultivated THP-1-derived macrophages, as is summarized in Figure 6. The induction of an apoA1-mediated cholesterol efflux does not only reflects cholesterol removal from macrophage foam cells; it is also an important initial step in the RCT process. This increases the de novo synthesis of nascent HDL or pre-β-HDL particles and leads to enhanced HDL function [38]. These effects of sitagliptin were observed in the presence, but not absence, of GLP-1, and were abolished by a GLP-1R antagonist, although not completely (Figure 5). A similar tendency was noted regarding the ABCA1 protein levels in the membrane fraction, as is shown in Figure 4A. These observations may indicate that sitagliptin phosphate can directly affect cholesterol efflux, suggesting that it upregulates apoA1-mediated cholesterol efflux via GLP-1R and that there may be additional direct pathways in macrophages.

Previous studies demonstrated GLP-1R expression in macrophages and THP-1 cells, as well as pancreatic β-cells [23]. Human primary monocytes and derived macrophages exhibit GLP-1R expression on their cell surfaces [33,39]. The presence of GLP-1R mRNA or protein has also been shown in macrophages derived from THP-1 cells [23,24], peritoneal macrophages from mice [23,40], and J774 cells [26]. Sitagliptin treatment has been previously reported in the plaque regression of apoE-deficient mice fed with a high-fat diet [24,25]. In addition, a clinical study has demonstrated that GLP-1 was associated with less cardiovascular disease [41]. These findings strongly suggest that, in addition to improving glucose metabolism, GLP-1R may contribute to anti-atherosclerotic effects in monocytes/macrophages. Recent research studies [24,42] have reported that (i) DPP-4 was present and tended to increase following differentiation in macrophages derived from THP-1 cells, though the presence of GLP-1R in THP-1 cells did not largely change between before and after differentiation, and (ii) sitagliptin demonstrated the role of intrinsic DPP-4 for THP-1 cell migration. Similarly, with the exception of the GLP-1 receptor pathway, this study revealed that sitagliptin might indicate that the increased apoA1-mediated cholesterol efflux through membrane ABCA1 upregulation is caused by the inhibition of intrinsic DPP-4 in THP-1 cells (Figure 6). However, the inhibition of GLP1 degradation by sitagliptin, which increased GLP concentrations, did not have a functionally equal power to apoA1 for cholesterol efflux (Figure 1A).

Many individuals with diabetes or insulin resistance have low-plasma HDL-cholesterol levels [10,11,43,44], and the mitigation of high glucose levels and insulin resistance often leads to increased plasma-HDL-cholesterol levels. Our results revealed that, aside from improving glucose metabolism, sitagliptin phosphate enhanced HDL function in cultivated macrophages in vitro, indicating that DPP-4 inhibition may directly affect macrophages via GLP-1R.

In vitro and vivo studies have shown that GLP-1 directly or indirectly increases ABCA1 protein expression in human-monocyte-derived macrophages and THP-1 cells [39,45], and regulates cholesterol homeostasis [46]. Conversely, in terms of cholesterol efflux, certain studies have shown that that the increased cholesterol efflux was independent of GLP-1, e.g., the effect of bariatric surgery on humans and rats [47]. In this study, we also found that GLP-1 directly upregulated ABCA1 protein levels in the membrane fraction of THP-1 cells but did not upregulate whole-cell expression. There was no change in the macrophage-membrane ABCA1 protein expression when using sitagliptin phosphate without GLP-1, indicating the need for the direct action of GLP-1. Furthermore, the DPP-4 inhibitor prevented GLP-1 degradation. ABCA1 is localized to the plasma membrane to ensure cholesterol homeostasis and plays a certain role in HDL production and cellular cholesterol efflux to outside. Therefore, it can be inferred that increased levels of ABCA1 in THP-1 cells, especially the membrane fraction, increase via the GLP-1 receptor pathway [3,45]. Therefore, it is indicated that the ability for increasing the ABCA1 of sitagliptin was weaker than that of GLP-1 in the GLP-1 receptor pathway, even if intrinsic DPP-4 demonstrated this effect on ABCA1 protein expression in the cell-membrane fraction of THP-1 cells (Figure 4). Our results may support previous findings from a clinical meta-analysis, demonstrating that GLP-1 agonists were superior to DPP-4 inhibitors in suppressing cardiovascular disease mortality [28]. However, the reason for the discrepancy in cholesterol efflux results between sitagliptin and vildagliptin was unclear in the present study.

cAMP and ligands of the nuclear transcription factors LXRα and RXRα have been identified as enhancers of ABCA1 gene expression, and cAMP has been shown to activate apoA1-mediated cellular cholesterol efflux [7,31,34,48]. It is well known that GLP-1R is a G-protein-coupled receptor, and its activation increases the intracellular cAMP levels via the activation of adenylate cyclase [49]. In contrast, ABCA1 activity is upregulated by cAMP via PKA phosphorylation [8,50]. In addition, GLP-1 was recently reported to increase cholesterol efflux in HepG2 [51] and glomerular endothelial cells [52], but not in macrophages. The latter study showed that the GLP-1R pathways, including cAMP/PKA activation, were involved in the upregulation of cholesterol efflux via ABCA1. Our observations also demonstrated that sitagliptin phosphate combined with GLP-1 significantly elevated apoA1-mediated cholesterol efflux in a dose-dependent manner. However, a PKA inhibitor was not found to affect apoA1-mediated cholesterol efflux in macrophages, indicating that pathways other than PKA–ABCA1 phosphorylation due to cAMP might be involved in ABCA1 upregulation in macrophages. The exchange protein directly activated by cAMP (EPAC) is known for intracellular cAMP receptors, along with PKA. EPAC1 is an isoform of EPAC which is expressed in many tissues and blood, including macrophages. Involvement of the cAMP/EPAC1 signaling pathway in the development of atherosclerosis via the increased lipid accumulation of macrophages was recently reported [53]. The outcome was thought to be in contrast to reduced lipid content by increased cholesterol efflux with sitagliptin. Therefore, cAMP/EPAC signaling may not be involved in the present study.

The addition of exendin-(9–39) inhibited the upregulation of apoA1-mediated cholesterol efflux (Figure 5) and membrane ABCA1 expression (Figure 4A) due to GLP-1 and sitagliptin phosphate administration, respectively. This indicated that both effects were mainly, but not exclusively, caused via the GLP-1R pathway. This is further supported by previous findings stating that incretin antagonists, a GLP-1, and a glucose-dependent insulinotropic polypeptide, could partially attenuate the suppressive effects of DPP-4 inhibitors on atherosclerosis in diabetic mice [26]. While the anti-atherogenic effects of DPP-4 inhibitors are mainly incretin-dependent, sitagliptin may affect cholesterol efflux and ABCA1 expression independently of the GLP-1R pathway because a GLP-1R antagonist did not completely cancel the effects on cholesterol efflux and ABCA1 expression. 

Our study has several limitations. First, we evaluated ABCA1 modification and degradation only in vitro. In addition, we only evaluated the effect of PKA, which is one of the factors suggested previously [54,55] on ABCA1 modification. ABCA1 protein degradation by calpain and the ubiquitin–proteasome system was previously reported, with a reported half-life that was relatively short (approximately 1–2 h) [35,56]. Finally, we did not measure the duration of drug action. 

These observations suggest that GLP-1 regulation by sitagliptin phosphate could be an attractive therapy for increasing apoA1-mediated cholesterol efflux via the upregulation of ABCA1 and the generation of HDL.

## 5. Conclusions

In conclusion, the present study revealed that sitagliptin elevates cholesterol efflux in cultivated macrophages and may exert anti-atherosclerotic effects that are independent of any improvements in glucose metabolism. Our results suggest that sitagliptin enhances HDL function by inducing de novo HDL synthesis.

## Figures and Tables

**Figure 1 biomolecules-13-00228-f001:**
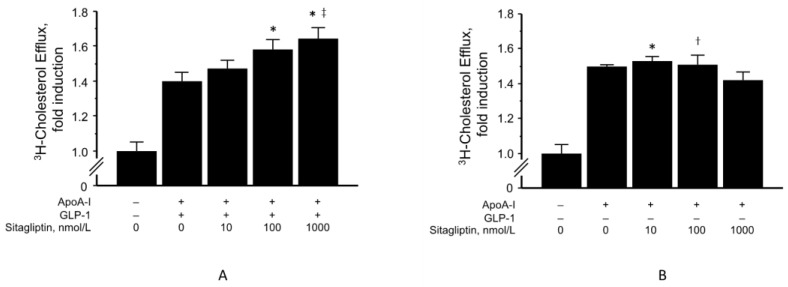
Effect of sitagliptin in the presence (**A**) or absence (**B**) of GLP-1 (10 nmol/L) on apoA1-mediated cholesterol efflux. Values represent mean ± SD. * *p* < 0.01; ^†^
*p* < 0.05 vs. apoA-I (–), ^‡^
*p* < 0.05 vs. apoA1 + GLP-1. Each group, *n* = 5–6.

**Figure 2 biomolecules-13-00228-f002:**
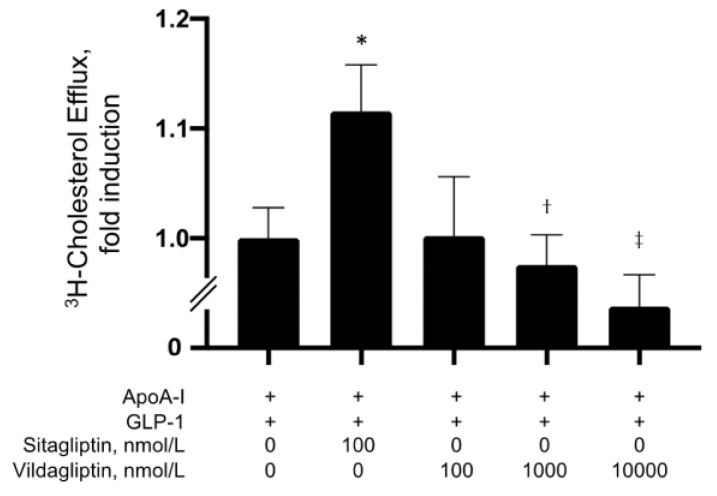
Effect of sitagliptin and vildagliptin in the presence of GLP-1 (10 nmol/L) on apoA1-mediated cholesterol efflux. Values represent mean ± SD. * *p* < 0.05 vs. apoA-I + GLP-1, ^†^
*p* < 0.05; ^‡^
*p* < 0.01 vs. sitagliptin (100 nmol/L). Each group, *n* = 4–8.

**Figure 3 biomolecules-13-00228-f003:**
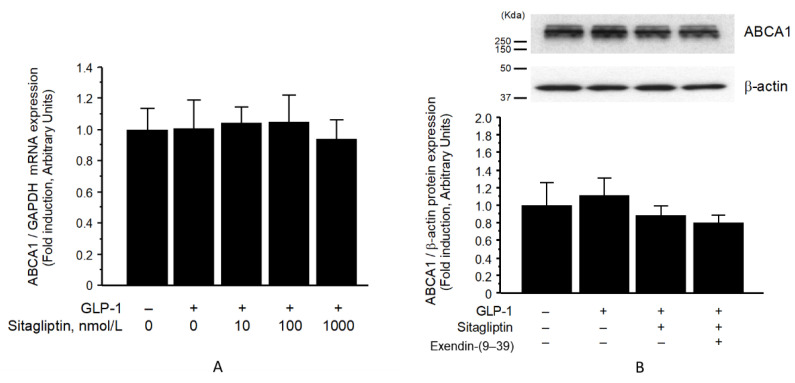
Effect of sitagliptin in the presence of GLP-1 (10 nmol/L) on ABCA1 mRNA (*n* = 8–12) (**A**) and protein (*n* = 5) (**B**). Expressions in whole-cell lysates of THP-1 cells. Values represent mean ± SD. Abbreviations: ABCA1, adenosine triphosphate-binding cassette transporter A1; GAPDH, glyceraldehyde-3-phosphate dehydrogenase.

**Figure 4 biomolecules-13-00228-f004:**
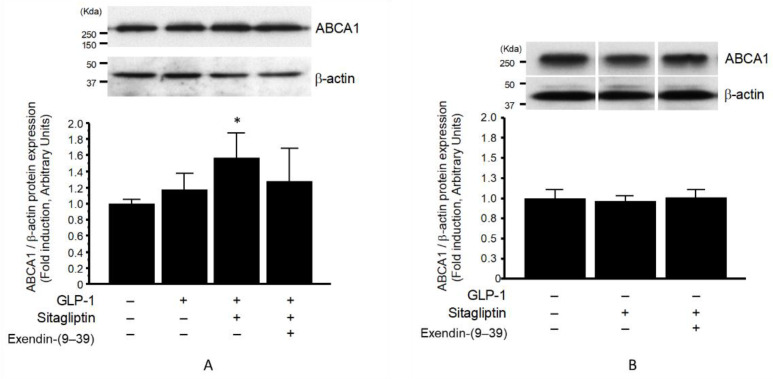
Effect of sitagliptin (1 μmol/L) in the presence (*n* = 5–8) (**A**) or absence (*n* = 3–4) (**B**) of GLP-1 (10 nmol/L) on ABCA1 protein expression in the cell membrane fraction of THP-1 cells. Values represent mean ± SD. * *p* < 0.05 vs. GLP-1 (–).

**Figure 5 biomolecules-13-00228-f005:**
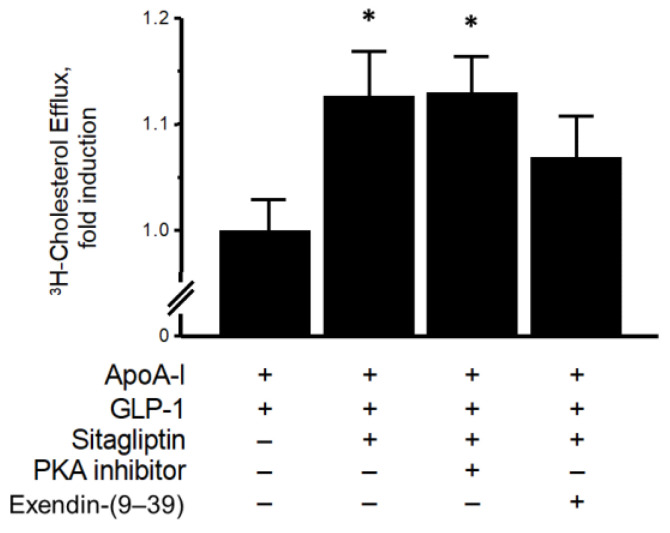
Influence of PKA inhibitor and exendin-(9–39) (150 nmol/L) on elevation of apoA1-mediated cholesterol efflux due to sitagliptin (1 μmol/L) in the presence of 10 nmol/L GLP-1. Values represent mean ± SD. * *p* < 0.01 vs. apoA1 + GLP-1. Each group, *n* = 5–6.

**Figure 6 biomolecules-13-00228-f006:**
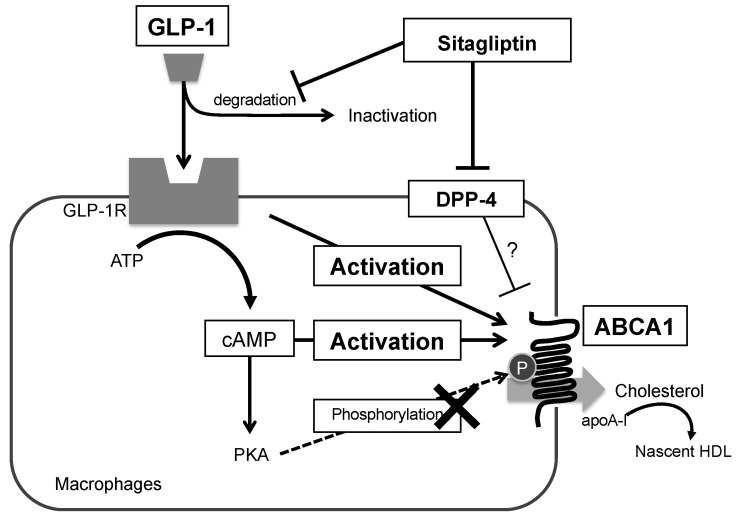
Graphical abstract for this study. Abbreviations: GLP-1, glucagon-like peptide-1; DPP-4, dipeptidyl peptidase-4; ABCA1, adenosine triphosphate-binding cassette transporter A1; PKA, protein kinase A; HDL, high-density lipoproteins.

## Data Availability

Not applicable.

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
