# Peer review of "Dipeptidyl Peptidase-4 Inhibitor Sitagliptin Phosphate Accelerates Cellular Cholesterol Efflux in THP-1 Cells"

_biomolecules, 2023, doi:10.3390/biom13020228_

Round 1
Reviewer 1 Report
General Comments
This is an in vitro study showing that the DPP4 inhibitor sitagliptin promotes cholesterol efflux from THP-1 cells via enhanced ABCA1 expression. This reviewer raises several concerns below.
Specific Comments
1.Reasons for choosing sitagliptin
The authors should describe in the Introduction why they chose sitagliptin among the many DPP4 inhibitors. In addition, since the GLP-1R antagonist only partially inhibited the action of sitagliptin, similar experiments should have been done with other DPP4 inhibitors.
2.DPP4 expression and secretion in THP-1 cells
The experimental results that sitagliptin promoted cholesterol efflux only in the presence of GLP-1 indicate that THP-1 cells must synthesize and secrete DPP4. The authors should state that there are findings or reports on this.
3.Validity of sitagliptin and GLP-1 concentrations
The authors should provide validation of the concentrations of sitagliptin and GLP-1 added to THP-1 cells.
4.Culture conditions for THP-1 macrophages
This paper is the result of a study on the anti-atherosclerotic effects of diabetes medications. Therefore, the authors should state whether the medium after treatment of THP-1 cells with PMA remains RPMI-1640 or is changed to DMEM. Also, if the medium was changed to DMEM, it should be clearly stated whether low glucose or high glucose medium was used.
5.Figure 3(B)
The authors should present experimental results in the presence of GLP-1.
6.Discussion on the localization of ABCA1
The effect of sitagliptin and GLP-1 on ABCA1 localization is not fully discussed. Why increase only ABCA1 in the plasma membrane fraction? The authors' speculation should be stated.
7.Post-translational regulation of ABCA1
The authors focused only on PKA, but ABCA1 is proteolytically degraded by calpain and the ubiquitin-proteasome system. There is a lack of experimentation and discussion regarding post-translational regulation of ABCA1. The authors should also describe the duration of action of sitagliptin and GLP-1. In addition, it would be useful to conduct experiments to observe ABCA1 proteolysis under conditions in which cycloheximide is used to stop the synthesis of new proteins.
8.Figure 5
Figure 5 is difficult to understand from the results of this study, in which PKA was not involved in sitagliptin's ABCA1 increase and promotion of cholesterol efflux.
Minor Comments
None.
Author Response
We attached files. Please see it.

Reviewer 2 Report
The DPPIV inhibitor sitagliptin, which increases intracellular cAMP by inhibiting GLP-1 decay, has been shown to increase macrophage cholesterol efflux by several unknown mechanisms. In fact, good results have been obtained in clinical trials in type 2 diabetic patients at high cardiovascular risk. Thus, the results of this study are indeed interesting. The reviewer requests some additions to improve this study.
1. The level of GLP1R expression in macrophages. Since GLP1R expression is primarily restricted to pancreatic beta cells, the reviewer wonders whether human macrophages, including THP-1 macrophages, really have a meaningful level of GLP1R gene expression. Please respond by gathering information from available gene expression databases and other sources. Also, is there any change in GLP1R expression in THP-1 cells after differentiation into macrophage-like cells?
2. The mechanism modulates ABCA1 expression. The authors suggest that cAMP upregulates macrophage cell surface ABCA1 by pathways other than PKA activation. I am very interested in this finding; is it possible to investigate the EPAC pathway?
Author Response
We attached files. Please see it.

Reviewer 3 Report
The authors provide initial evidence that increases of GLP1 by inhibition of dipeptidyl peptidase 4 with sitagliptin promotes cholesterol efflux from macrophages. THe data are innteresting and of potenial clinical relevance. However, there is some need to foster the conclusions.
1. As usual in preclinical studies, the number of data per experiment are rather low. THerefore non-parametric tests are to be preferred over parametric tests. At least the authors must show normal Gaussian frequency distribution of data before using parametric tests.
2. The experiment shown in figure 1 A will benefit from the demonstration of data on the effect of apoA-I in the absence of GLP1. THis data are shown in figure 1B and it seems that the addtion of GLP1 does not increase cholesterol efflux beyond apoA-I. Why then does inhibition of GLP1 degradation by sitagliptin have an effect? A dose response analysis of GLP1 effects will be helpful to clarify this discrepancy.
3. The absent effect of GLP1 on ABCA1 expression is clear from figure 2. However, the effect on ABCA1 protein expression is less clear. First the western blot does not prove the claim because it shows a decrease in beta-actin rather than an increase in ABCA1. Seconde the method described does not allow differentiation between cell surface and the total protein. TO this end cell surface biotinylation experiments must be done. Then both the lysate and the surface fractions must be shown separately with according independent protein markers (e.g. Tata binding protein for the lysate and Na/K- ATPase for the plasma membrane fraction). The beta-actin used by the authors does not allow any differentiation of total cell from cell surface.
4.THe authors tested apoA-I mediated cholesterol efflux. Is also HDL-mediated cholesterol efflux changed?
5, The study will benefit from the verificaiton of the most important findings by the use of another dipeptidyl peptidase inhibitor (e.g) and a GLp1 receptor agonist (eg. liraglutide) . In the same line it will be interesting to reproduce the effects in primary monocyte derived macrophages.
6. THe authors did a rather comprehensive discussion of the literature in the context of their findings. ONe study will be interesting to be included: (PMID 25673670). In this study, bariatric surgery improved several HDL functions including cholesterol efflux from macrophages. However, by contrast to improvements of other HDL functions notably on the endothelium, the increased cholesterol efflux was found to be independent of GLP1. THis data are somehow in contrast to the data presented here and deserve discussion
Author Response
We attached files. Please see it.

Round 2
Reviewer 1 Report
This reviewer has no further comments.
Reviewer 3 Report
the authors addressed my comments. They revised the manuscript when no additional experiments were needed. The manuscript would be improved by addtional data on cell surface ABCA1, HDL-mediated efflux (whicjh also involves ABCA1 in addtion to other inducers), and primary monocytes. But I understand that the time constraints of the editors do not allow this